# Intestinal delta-6-desaturase activity determines host range for *Toxoplasma* sexual reproduction

**Bruno Martorelli Di Genova**[1], **Sarah K. Wilson**[1], **J. P. Dubey**[2], **Laura J. Knoll**[1]*

**1** Department of Medical Microbiology and Immunology, University of Wisconsin–Madison, Madison, Wisconsin, United States of America, **2** Animal Parasitic Diseases Laboratory, Animal and Natural Resources Institute, United States Department of Agriculture, Agricultural Research Service, Beltsville, Maryland, United States of America

* ljknoll@wisc.edu

**Data Availability Statement:** All relevant data are within the paper and its Supporting Information files.

**Funding:** This research was supported by the National Institutes of Health (NIH) National

## Abstract

Many eukaryotic microbes have complex life cycles that include both sexual and asexual phases with strict species specificity. Whereas the asexual cycle of the protistan parasite *Toxoplasma gondii* can occur in any warm-blooded mammal, the sexual cycle is restricted to the feline intestine. The molecular determinants that identify cats as the definitive host for *T. gondii* are unknown. Here, we defined the mechanism of species specificity for *T. gondii* sexual development and break the species barrier to allow the sexual cycle to occur in mice. We determined that *T. gondii* sexual development occurs when cultured feline intestinal epithelial cells are supplemented with linoleic acid. Felines are the only mammals that lack delta-6-desaturase activity in their intestines, which is required for linoleic acid metabolism, resulting in systemic excess of linoleic acid. We found that inhibition of murine delta-6-desaturase and supplementation of their diet with linoleic acid allowed *T. gondii* sexual development in mice. This mechanism of species specificity is the first defined for a parasite sexual cycle. This work highlights how host diet and metabolism shape coevolution with microbes. The key to unlocking the species boundaries for other eukaryotic microbes may also rely on the lipid composition of their environments as we see increasing evidence for the importance of host lipid metabolism during parasitic lifecycles. Pregnant women are advised against handling cat litter, as maternal infection with *T. gondii* can be transmitted to the fetus with potentially lethal outcomes. Knowing the molecular components that create a conducive environment for *T. gondii* sexual reproduction will allow for development of therapeutics that prevent shedding of *T. gondii* parasites. Finally, given the current reliance on companion animals to study *T. gondii* sexual development, this work will allow the *T. gondii* field to use of alternative models in future studies.

Research Service Award T32 AI007414 (SKW), R01AI144016-01, 5R21AI1123289-02, and 5R03AI104697-02 (LJK) and the Morgridge Metabolism Interdisciplinary Fellowship from the Morgridge Institute for Research (BMDG). The funders had no role in study design, data collection and analysis, decision to publish, or preparation of the manuscript.

**Competing interests:** The authors have declared that no competing interests exist.

**Abbreviations:** AO2, amine oxidase, copper-containing protein 2; BRP1, bradyzoite rhoptry protein-1; DBA, *Dolichos biflorus* agglutinin; GRA11B, dense granule protein 11B; MOI, multiplicity of infection; qPCR, quantitative PCR; RNAseq, RNA sequencing; SAG1, surface antigen 1; TUB1A, tubulin 1A; ZO-1, zona occludens-1.

## Introduction

The apicomplexan parasite *Toxoplasma gondii* causes a chronic infection in nearly one-third of the human population and is well known for causing congenital infections leading to blindness, mental retardation, and hydrocephaly of the developing fetus. *T. gondii* has a complex life cycle containing both sexual and asexual phases. The *T. gondii* asexual cycle can occur in any warm-blooded animal when contaminated food or water is consumed and *T. gondii* disseminates throughout the host, converting to an encysted form in muscle and brain tissue. In contrast, the *T. gondii* sexual cycle is restricted to the feline intestinal epithelium, culminating in the excretion of environmentally resistant oocysts [1]. The molecular basis for this species specificity is unknown.

During feline infection, the ingested bradyzoites are released by pepsin and acid digestion in the stomach. Bradyzoites then invade the feline intestinal epithelium and differentiate into five morphologically distinct types of schizonts [2]. Within 2 days in the feline intestine, parasites progress through all five stages of schizonts and then develop into merozoites. The merozoites undergo a limited proliferation of two to four doublings before they differentiate into macrogametes and microgametes. The macro- and microgametes fuse to produce diploid oocysts, which develop thick impermeable walls and are shed in the feces. Cats usually excrete 2–20 million oocysts per day in their feces and usually shed oocysts 5–10 days post infection [3,4]. In ambient air and temperature, oocysts undergo a sporulation process to mature and become infectious. Both mitosis and meiosis occur during sporulation to produce eight haploid sporozoites encased within the oocyst wall. *T. gondii* oocysts are stable for 18 months in unfavorable environmental conditions and resistant to many chemical disinfectants [5].

## Results and discussion

### Linoleic acid is critical for sexual development in cultured cat cells

To determine the molecular mechanisms that define the species specificity of *T. gondii* sexual development, we generated cat intestinal organoids (Fig 1a) and then seeded these epithelial cells onto glass coverslips. These monolayers displayed intestinal epithelial properties, including polarization and tight junction formation (Fig 1b). To simulate natural infection, *T. gondii* was harvested from mouse brains 28–40 days after primary infection, and the parasites were released from the brain cysts by pepsin and acid digestion. After neutralization with sodium carbonate, parasites were seeded onto the cat intestinal monolayers, incubated for 5 days, and stained for markers of the parasite presexual stage called a merozoite [6,7]. Although we observed occasional dense granule protein 11B (GRA11B) and bradyzoite rhoptry protein-1 (BRP1) staining, the vast majority of the culture was negative for these merozoite markers (Fig 1c), suggesting that a required nutrient was limiting under these culture conditions. Because recent studies showed that the *T. gondii* asexual stages scavenge fatty acids, particularly oleic acid, from the host [8] and that sexual development of many fungi is dependent on linoleic acid [9], we surmised that supplementation with these fatty acids could facilitate *T. gondii* sexual development. We added 200 μM oleic or linoleic acid to cat intestinal monolayer culture medium 24 hours prior to infection with *T. gondii*. After 5 days of infection, we found that the addition of linoleic acid but not oleic acid caused approximately 35% of the *T. gondii* to express both merozoite stage markers (Figs 1d–1g and 2a, S1 Data). Similarly, GRA11B mRNA was significantly more abundant in cat intestinal cells supplemented with linoleic acid compared to any other condition (Fig 2b, S2 Data). As seen in vivo cat intestine, GRA11B changes localization from within the parasite dense granule organelles in the early stages of development to the parasitophorous vacuole and parasitophorous vacuole membrane in later stages of

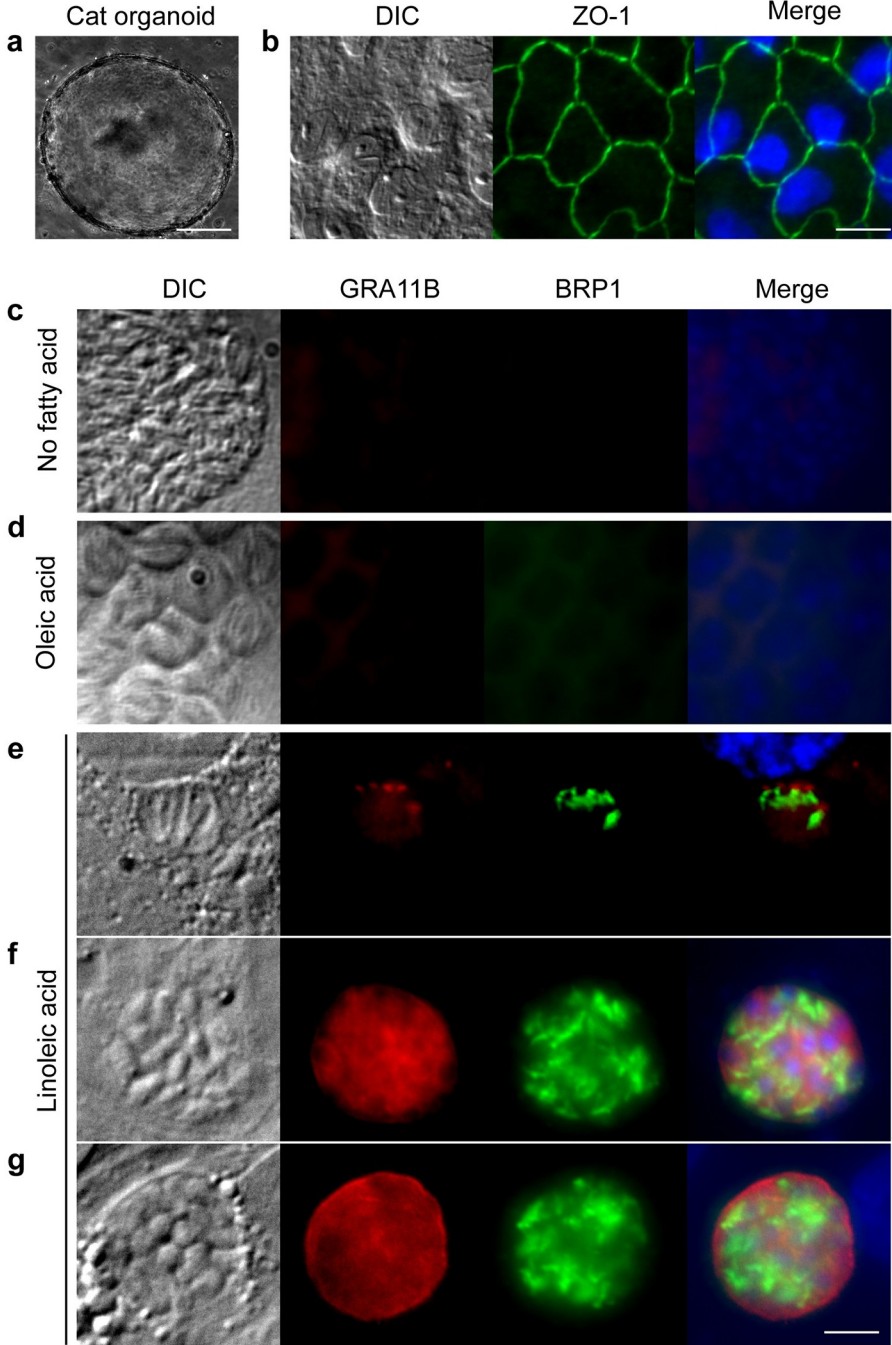

**Fig 1. Linoleic acid enhances progression through the sexual stages.** (a) Cat intestinal organoids were generated from small-intestine sections and were grown in basement membrane matrix. Example of a growing organoid, 100-μm size bar. (b) Intestinal organoids were dissociated using trypsin and single cells seeded onto glass coverslips to grow as monolayers. The cells in the monolayer expressed the tight junction protein ZO-1 (green), 20-μm size bar. Cat intestinal monolayers were incubated with either (c) no fatty acid supplementation, (d) 200 μM oleic acid, or (e, f, g) 200 μM linoleic acid for 24 hours and then infected with ME49 bradyzoites for 5 days. Parasites undergoing presexual development were commonly seen only with linoleic acid supplementation as marked by staining with GRA11B (red) or BRP1 (green). Parasites in (e) early, (f) middle, or (g) late stages of sexual development were noted by differential localization of GRA11B. All panels are 20 μm$^2$ with a 5-μm white size bar in the lower right corner. BRP1, bradyzoite rhoptry protein-1; DIC, differential interference contrast; GRA11B, dense granule protein 11B; ZO-1, zona occludens 1.

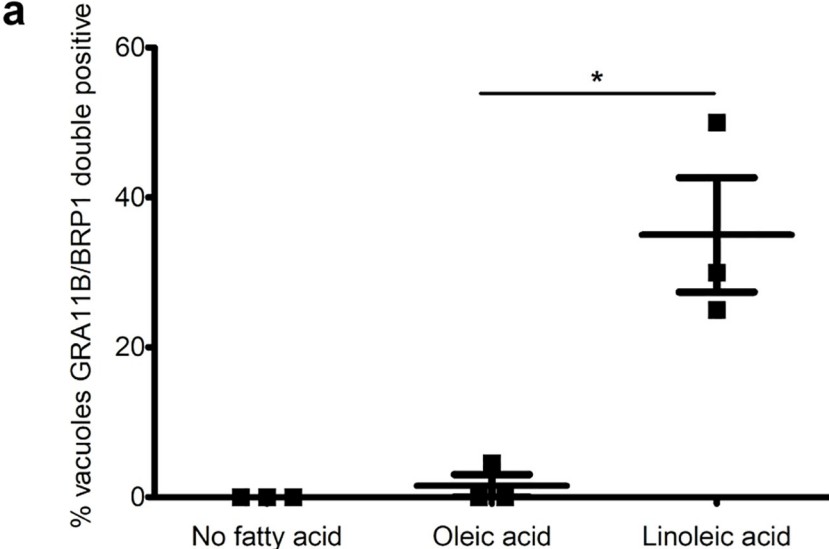

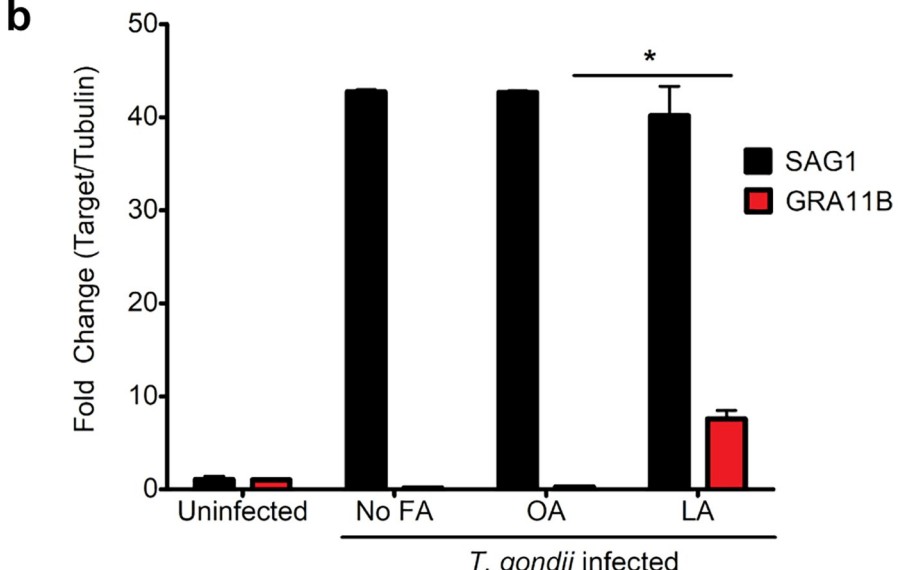

**Fig 2. Quantification of merozoites in cat tissue culture.** (a) Cat intestinal organoids were disassociated by trypsin and then grown as monolayers on glass slides. Slides were divided into three different groups: not supplemented with FA, supplemented with 200 μM OA, or supplemented with 200 μM LA. Monolayers were infected with *T. gondii* ME49 bradyzoites purified from brains of chronic infected mice at a 1:10 MOI. Five days after infection, staining for GRA11B and BRP1 along with DAPI allowed the percentage of vacuoles positive for GRA11B and BRP1 out of the total vacuoles to be determined. Total number of parasitophorous vacuoles were counted by positive DAPI staining and confirmed by morphology with DIC. Three biological replicates were counted, and on average, 35% of the total vacuoles were positive for both GRA11B and BRP1 in the LA-supplemented monolayers. *p-value = 0.0126 with $N$ = 3 by two-tailed unpaired *t* test. Straining for both BRP1 and GRA11B was used to ensure that merozoite stages were counted. RNAseq and immunofluorescent imaging of the cat intestinal epithelium show that GRA11B is exclusively expressed in merozoites [6]. BRP1 is a rhoptry protein that was initially found in bradyzoites; however, it is also expressed in merozoites [7]. (b) Cat intestinal monolayers were grown as described in (a), except monolayers were quenched by TRIzol 5 days post infection, RNA was extracted, and cDNA was synthesized using an oligo (dT) primer to amplify mRNA. Expression of SAG1 and GRA11B were quantified by qPCR and the fold change calculated in comparison with uninfected cells. TUB1A was used to normalize gene expression across samples. GRA11B expression was significantly more abundant in the LA-supplemented monolayers with two biological replicates. *p-value = 0.0155 with $N$ = 2 by two-tailed unpaired *t* test. BRP1, bradyzoite rhoptry protein-1; DIC, differential interference contrast; FA, fatty acid;

GRA11B, dense granule protein 11B; LA, linoleic acid; MOI, multiplicity of infection; OA, oleic acid; qPCR, quantitative PCR; RNAseq, RNA sequencing; TUB1A, tubulin 1A; SAG1, surface antigen 1.

development [6]. We see similar localization of GRA11B depending on vacuole size, likely representing early, middle, and late stages (Fig 1e–1g). Likewise, BRP1 has a localization similar to that previously seen in the rhoptry organelles in the apical end of the merozoite [7] (Fig 1e–1g).

Within the feline intestine, merozoites are known to differentiate into micro- and macrogametes that fuse to become diploid oocysts. After 7 days of infection, we saw round structures with reactivity to the macrogamete protein amine oxidase, copper-containing protein 2 (AO2) [10] in cat intestinal monolayers cultured with 200 μM linoleic acid but not in unsupplemented or oleic acid–supplemented cultures (Fig 3a–3c). PCR of these day 7 linoleic acid–supplemented cultures amplified message for AO2 as well as the predicted microgamete flagellar dynein motor protein TGME49_306338 with 44% identity to the homologue from the motile green alga *Chlamydomonas reinhardtii* (Fig 3d). In parallel, we assessed for the presence of intracellular oocyst wall biogenesis in these linoleic acid–supplemented cat cells by using the 3G4 antibody [11] that recognizes the *T. gondii* oocyst wall. There were approximately nine oocyst walls per cm$^2$ of cultured cat cells supplemented with 200 μM linoleic acid but none in not-supplemented or oleic acid–supplemented cultures (Fig 3e–3h, S3 Data). Addition of 20 μM linoleic acid did not enhance oocyst wall production, indicating that the concentration of linoleic acid was critical for proper development.

## Inhibition of delta-6-desaturase causes sexual development in cell culture mouse cells

The dependence of *T. gondii* sexual development on high levels of linoleic acid was intriguing because cats are the only mammal known to lack delta-6-desaturase activity in their small intestines [13,14]. Delta-6-desaturase is the first and rate-limiting step for the conversion of linoleic acid to arachidonic acid. Linoleic acid is the dominant fatty acid in cat serum, comprising 25%–46% of the total fatty acid [15–18], whereas rodents serum contains only 3%–10% linoleic acid [19–22]. We hypothesized that the lack of delta-6-desaturase activity in the cat small intestine allows for a buildup of linoleic acid from the diet, which then acts as a positive signal for *T. gondii* sexual development. To test this hypothesis, we infected mouse intestinal monolayers with *T. gondii* and supplemented them with linoleic acid and the chemical SC-26196, a specific inhibitor of the delta-6-desaturase enzyme, to establish high steady-state levels of linoleic acid [23]. Five days after infection of the mouse culture with *T. gondii*, we assessed merozoite markers BRP1 [7] and GRA11B [6]. We detected expression of GRA11B and BRP1 in mouse intestinal cells only when supplemented with both linoleic acid and SC-26196 (Fig 4). These data suggest that the delta-6-desaturase enzyme must be inhibited in order for high enough levels of exogenous linoleic acid to increase and induce *T. gondii* sexual development in nonfeline intestinal cells. Similar to cat cells, mouse intestinal monolayers supplemented with both linoleic acid and SC-26196 had approximately 26% of the *T. gondii* vacuoles expressing both BRP1 and GRA11B (S1 Fig, S4 Data).

## Inhibition of delta-6-desaturase causes sexual development in live mice

Oocysts excreted in cat feces must undergo a sporulation process to become infectious to the next host. We attempted to sporulate the round structures containing oocyst wall antigen that were derived from either cat or inhibited mouse cultured intestinal cells at room temperature

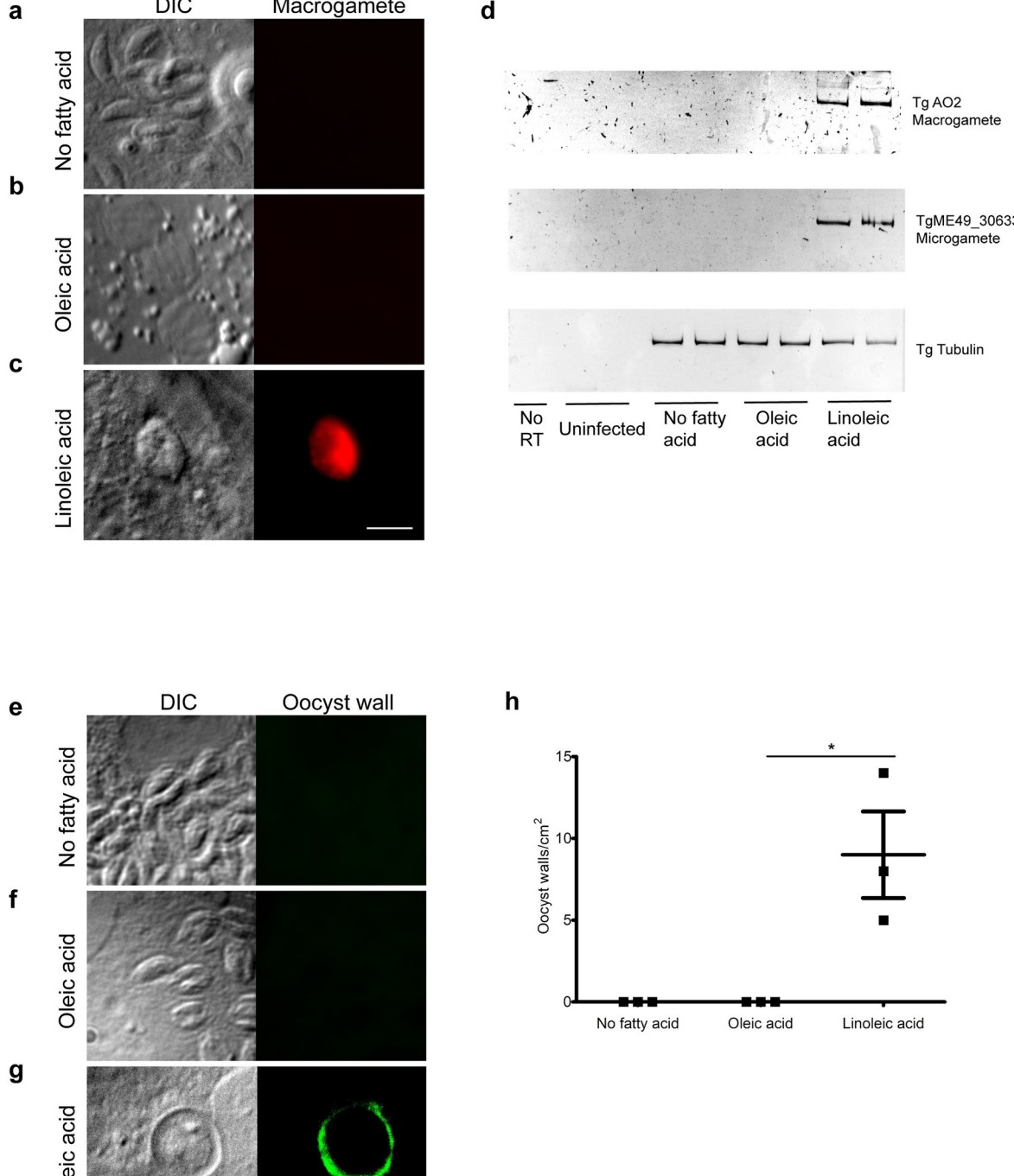

**Fig 3. Identification of gametes and intracellular oocysts in cat tissue culture.** Cat intestinal organoids were disassociated by trypsin and then grown as monolayers on glass slides. Monolayers were grown to confluency and then were incubated with either no fatty acid supplementation (a and e), 200 μM oleic acid (b and f), or 200 μM linoleic acid (c and g) for 24 hours and infected with ME49 bradyzoites purified from brains of chronic infected mice. After 7 days, monolayers were incubated with mouse anti-AO2 (a–c) or mouse monoclonal IgM 3G4 (e–g). The amiloride-sensitive AO2 is an enzyme exclusively expressed in macrogametes and early oocysts and has a possible role in oocyst wall biogenesis [10]. AO2 expression was only detected by immunofluorescence in the monolayers supplemented with linoleic acid (c). 3G4 is a mouse monoclonal antibody produced by immunizing mice with purified oocyst walls [11]; thus, it is a marker of oocyst wall biogenesis. Only monolayers supplemented with linoleic acid (g) had positive 3G4 vacuoles. All panels are 20 μm² with a 5-μm white size bar in the lower right corner. (d) Markers for macrogamete and microgamete expression were also evaluated by PCR. Cat intestinal monolayers

were grown in 24-well plates until confluency and then infected with *T. gondii* bradyzoites in duplicate using the same conditions as above. Seven days post infection, RNA was extracted with TRIzol, and cDNA was synthesized using an oligo (dT) primer to only amplify mRNA. AO2 was again used as a marker for macrogametes, and the expected PCR product is 218 bp. To assess microgamete presence, we selected the gene TgME49_306338, which is overexpressed in the gametes stage, corresponded to day 7 post infection in cats [12], and has 44% identity to a protein expressed in the flagella of the motile green algae *C. reinhardtii*. The expected PCR product for TgME49_306338 is 160 bp. TUB1A was used as an input control and results in a 172-bp product. "No RT" corresponds to a cDNA synthesis reaction without the addition of RT as a control for genomic DNA contamination. Equivalent amounts of cDNA per sample were used as a template for each PCR reaction, and the products were separated on an acrylamide gel. Bands with the correct size showing AO2 and TgME49_306338 expression were only observed in linoleic acid–supplemented monolayers. (h) The number of positive oocyst walls stained with 3G4 were quantified. Cat intestinal monolayers were infected with *T. gondii* bradyzoites and after 7 days, fixed with 3.7% formaldehyde in PBS, and incubated with 3G4 as showed in (e, f, and g). The number of positive oocyst walls were counted in each slide and divided by the area of slide in $cm^2$. The number of positive oocysts walls in monolayers supplemented with linoleic acid was significantly higher than those supplemented with oleic acid in three biological replicates. *$p$-value = 0.0272 with $N$ = 3 by two-tailed unpaired $t$ test. AO2, amine oxidase, copper-containing protein 2; DIC, differential interference contrast; IgM, immunoglobulin M; RT, reverse transcriptase; TUB1A, tubulin 1A.

with aerosolization for 7–14 days. Unfortunately, few structures were obtained from the monolayers, they did not appear to sporulate, and they were not infectious to mice. We hypothesized that *T. gondii* oocyst development and infectivity would require physiological conditions in a whole animal that could not be recapitulated in tissue culture. To test this hypothesis, we inhibited delta-6-desaturase activity in the intestines of live mice. The delta-6-desaturase inhibitor SC-26196 is effective as an anti-inflammatory agent in whole-animal experiments [24]. Because it was previously seen that sporozoites shifted to the rapidly replicating asexual stage called a tachyzoite, within 8 hours after the oral inoculation into rats [25] we fed the mice a linoleic acid–rich diet and pretreated them with the delta-6-desaturase inhibitor SC-26196 (or a no-inhibitor control) 12 hours prior to oral infection with *T. gondii* and every 12 hours thereafter. In mice fed both the linoleic acid–rich diet and the SC-26196 inhibitor, 7 days after infection, quantitative PCR (qPCR) of ileum cDNA showed high expression of the *T. gondii* merozoite marker GRA11B and low expression of the asexual tachyzoite surface antigen 1 (SAG1) [26] (Fig 5a, S5 Data). Ileum sections on day 7 post infection were paraffin-embedded and stained with hematoxylin and eosin or reticulin stain. Presexual and early oocysts stages were present only in the tissue of mice fed linoleic acid and the delta-6-desaturase inhibitor (Fig 5b and 5c).

As early as day 6 post infection, oocyst-like structures showing 3G4 antibody-positive staining were present in the mouse feces (Fig 5d) and increased in number until day 7, when the mice were euthanized. qPCR on genomic DNA from mouse fecal samples showed that *T. gondii* genomic DNA was detectable only in mice treated with SC-26196 (Fig 6a, S6 Data), indicating that delta-6-desaturase must be inactivated in mice for *T. gondii* sexual stages to develop in the mouse gut. Mice produced 1,000–10,000 oocysts/gram dry feces. To increase the number and duration of oocysts shedding, mice were fed the SC-26196 inhibitor every 12 hours only until day 5 post infection. Oocysts were monitored in the feces, with the peak of shedding being days 8–9 with between 100,000–150,000 oocysts/gram dry feces (Fig 6b, S7 Data), which is within the range seen for cats (2,000–1,500,000 oocysts/gram of feces [3,4]).

## Mouse-derived oocysts sporulate and are infectious

*T. gondii* oocysts are susceptible to desiccation, making them unable to sporulate [29]. Therefore, the mouse feces or the intestinal contents were immediately placed in saline and sporulated at room temperature with aerosolization. Cat-derived oocysts are usually stable after a 30-minute incubation in undiluted bleach (5% sodium hypochlorite) and long-term incubation in 2% sulfuric acid [5]. Because mouse-derived oocysts were not as resistant to bleach and 2% sulfuric acid as cat-derived oocysts, they were sporulated in saline with antibiotics. After 7

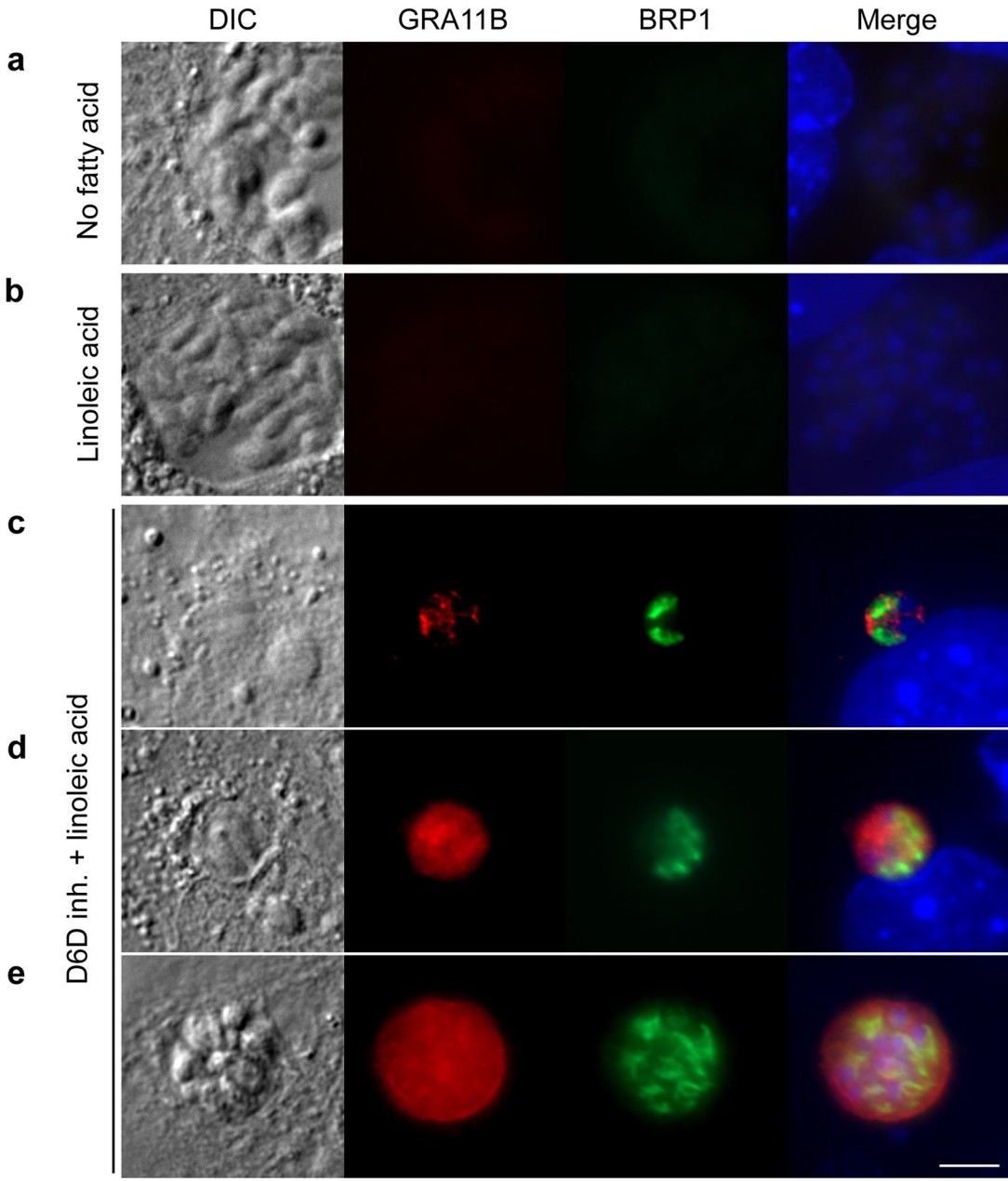

**Fig 4. Inhibition of delta-6-desaturase permits sexual development in mouse culture.** Mouse intestinal monolayers were incubated with either (a) no fatty acid supplementation, (b) 200 μM linoleic acid, or (c, d, e) 200 μM linoleic acid plus the delta-6-desaturase inhibitor ("D6D inh.") SC-26196 for 24 hours and then infected with ME49 bradyzoites for 5 days. Only in cultures supplemented with linoleic acid and SC-26196 were parasites undergoing presexual development detected by staining with GRA11B (red) or BRP1 (green). Parasites in (c) early, (d) middle, or (e) late stages of development were noted by differential localization of GRA11B. All panels are 20 μm$^2$ with a 5-μm white size bar in the lower right corner. BRP1, bradyzoite rhoptry protein-1; DIC, differential interference contrast; GRA11B, dense granule protein 11B.

days, sporulation was evident in approximately 50% of the oocysts by visualization of sporozoites, a deep blue autofluorescent wall [30] (Fig 6c), and reactivity with the 4B6 antibody that recognizes the two individual sporocysts within the oocysts [11] (Fig 6d). The sporulated oocysts were infectious to mice, as seen by serum conversion (S2 Fig) and cysts in the brains

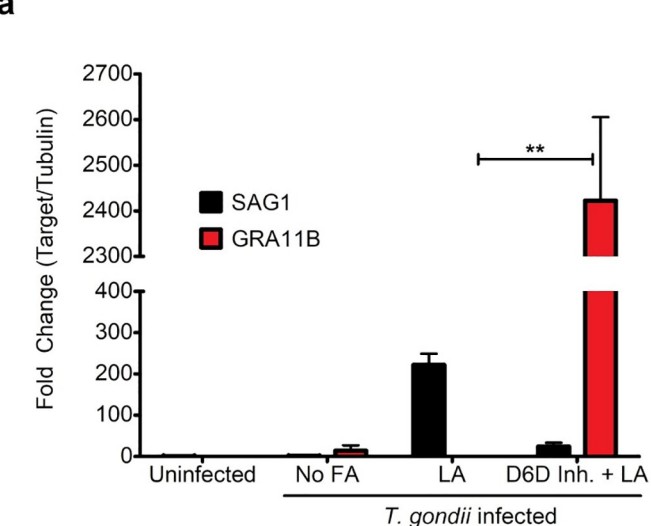

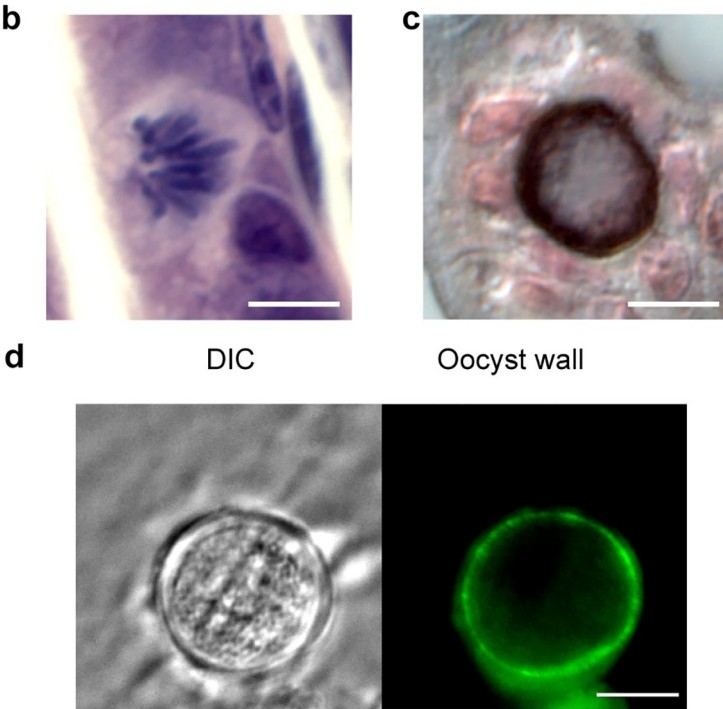

**Fig 5. Mice shed oocysts after inhibition of D6D.** Mice were gavage fed LA and the D6D inhibitor ("Inh.") SC-26196 12 hours prior to infection with ME49 bradyzoites and then every 12 hours for the 7 days of infection. (a) qPCR of cDNA from the ileum for tachyzoite marker SAG1 (black) and GRA11B (red) shows that GRA11B is significantly up-regulated only in the presence of SC-26196 ($p$-value = 0.0057 with $N$ = 2 by two-tailed unpaired $t$ test). (b) Ileum sections on day 7 post infection were paraffin-embedded and stained with hematoxylin and eosin to visualize presexual stages. (c) Early intracellular oocysts were observed in ileums of D6D inhibited mice impregnated with silver (reticulin stain) and photographed using differential contrast imaging to delineate the oocyst wall in dark brown [27]; 10-μm black size bar in the lower right corner. (d) Fresh oocysts were fixed in 3.7% formaldehyde in suspension, incubated with mouse monoclonal antibody 3G4 overnight, and then incubated with goat anti-mouse Alexa Fluor 488 secondary antibody. Panels are 20 μm$^2$ with a 5-μm white size bar in the lower right corner. D6D, delta-6-desaturase; DIC, differential interference contrast; FA, fatty acid; GRA11B, dense granule protein 11B; LA, linoleic acid; qPCR, quantitative PCR; SAG1, surface antigen 1.

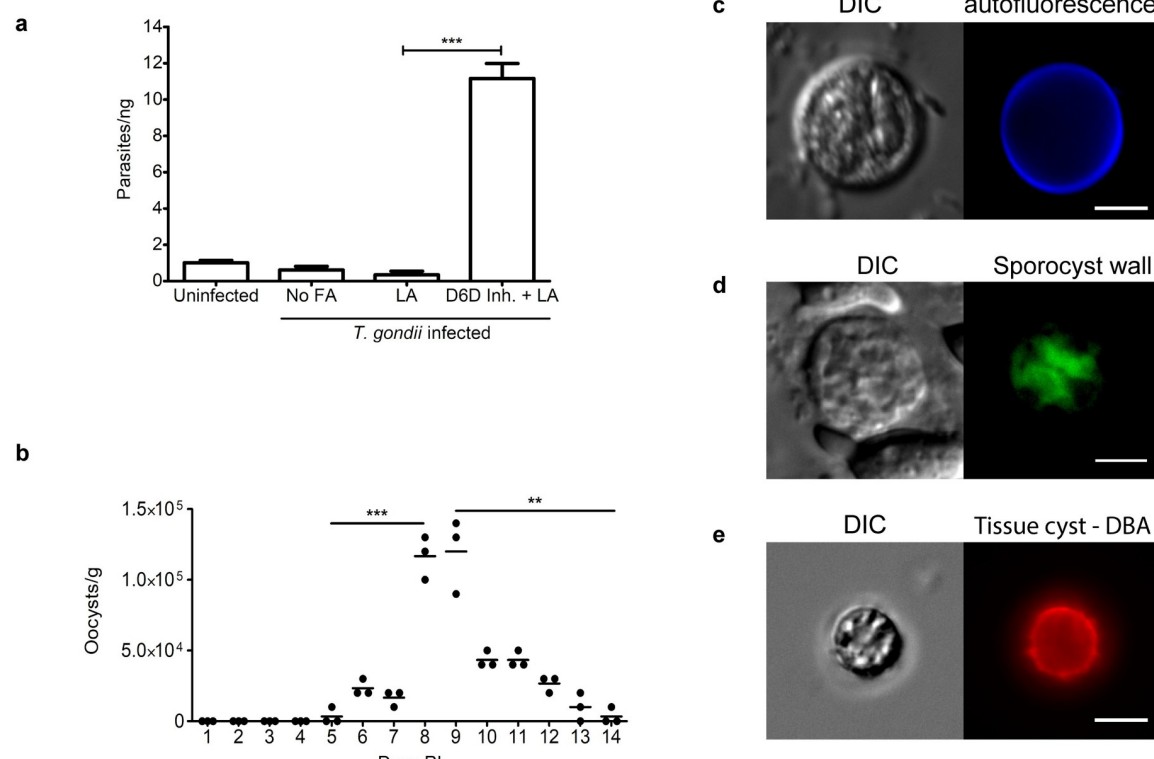

**Fig 6. Mouse oocysts are infectious.** (a) qPCR on genomic DNA from mouse fecal samples shows that *T. gondii* genomic DNA is detected only in mice treated with SC-26196 (*p*-value = 0.0002 with *N* = 3 by two-tailed unpaired *t* test). (b) Counts of the number of oocysts/gram of feces over time. *** *p*-value = 0.0003 day 5 versus 8 and ** *p*-value = 0.0017 day 9 versus 14. (c) After 7 days in sporulation conditions, sporocysts were visible by DIC, and blue autofluorescence of the oocyst walls was enhanced. All panels are 20 μm² with a 5-μm white size bar in the lower right corner. (d) To expose the sporocyst wall to the 4B6 antibody [11], the sporulated oocysts were dried to the slides, fixed and permeabilized with cold acetone for 30 minutes, incubated with mouse monoclonal antibody 4B6 overnight, and then incubated with goat anti-mouse Alexa Fluor 488 secondary antibody. (e) At 28 days PI with oocysts, *T. gondii* cysts were purified from the brains of mice and detected by DBA [28] (red). All panels are 20 μm² with a 5-μm white size bar in the lower right corner. D6D Inh., delta-6-desaturase inhibitor; DBA, *Dolichos biflorus* agglutinin; DIC, differential interference contrast; FA, fatty acid; LA, linoleic acid; PI, post infection; qPCR, quantitative PCR.

28 days later (Fig 6e, S3 Fig). Similar to oocysts derived from a cat, these mouse-derived sporulated oocysts were stable and infectious for at least 3 months when stored at 4 ˚C.

## Concluding remarks

All together, these results define the mechanism of species specificity for *T. gondii* sexual development and show that the species barrier can be broken for *T. gondii* sexual development by inhibiting delta-6-desaturase activity in the intestines of a nonfeline host. The lack of delta-6-desaturase activity and the buildup of linoleic acid likely enhance *T. gondii* sexual development in multiple ways. First, prior work suggests linoleic acid is cytotoxic for the asexual tachyzoite stage [31]; thus, tachyzoite development would be halted in a linoleic-rich environment. Second, inhibition of delta-6-desaturase likely lowers arachidonic acid levels, which would alter the production of immune lipid mediators known as eicosanoids. Finally, the dramatic difference between oleic acid with one double bond and linoleic acid with two highlights that linoleic acid is probably used as a signaling molecule and not to meet basic nutritional needs. Quorum-sensing for sexual reproduction in fungi is dependent on oxygenation of linoleic acid but not oleic acid [9]. The multiple host and *T. gondii* cyclooxygenases and

lipoxygenases might oxygenate linoleic acid to an oxylipin signaling molecule for *T. gondii* sexual development to proceed. Other protozoan parasites also rely on host derived lipids, which may be used as signaling molecules as well as lipid sources for parasite growth during their life cycles [32,33].

## Materials and methods

### Ethics statement

Mice were treated in compliance with the guidelines set by the Institutional Animal Care and Use Committee (IACUC) of the University of Wisconsin School of Medicine and Public Health (protocol #M005217). Cats were treated in compliance with the guidelines set by the IACUC of the United States Department of Agriculture, Beltsville Area (protocol #15–017). Both institutions adhere to the regulations and guidelines set by the National Research Council.

### Intestinal organoids

Cat intestinal organoids were established from jejunum sections obtained from fetal small-intestinal sections. Mouse intestinal organoids were established from jejunum sections from 8-week-old C57BL/6J male mice. Organoids were generated as described previously [34]. Briefly, intestinal sections were washed in ice-cold PBS containing 0.1 mg/mL streptomycin and 100 U/mL penicillin for 20 minutes. Sequentially, EDTA (Sigma) was added to a final concentration of 2 mM and the tissue incubated for 40 minutes at 4 ˚C. The tissue was then rinsed in cold PBS without EDTA and vigorously shaken until crypts were seen in the supernatant. The crypt suspension was filtered using a 70-μm cell strainer, and the crypts were centrifuged at 80*g* for 5 minutes. The cells were resuspended in Matrigel (BD Biosciences), pipetted into a 24-well plate, allowed to polymerize, and then covered with organoid medium. The organoid medium contains Advanced DMEM/F12 with 2 mM Glutamax, 20 mM HEPES, 1 × B27, 1 × N2, 10% v/v fetal bovine serum, 10 mg/L insulin, 5.5 mg/L transferrin, 0.67 mg/L selenite, penicillin and streptomycin (all from Invitrogen), 50 ng/ml human EGF (R&D systems), 10 mM nicotinamide (Sigma), 3 μM CHIR99021 and 10 μM Y-27632 (both Selleckchem), and 50% v/v conditioned medium obtained from L-WRN cell line (ATCC CRL 3276). The medium was changed every other day, and the organoids were expanded by passing the cells through a 25-gauge needle every week. All experiments were done with cells at passage 2 to 5, and cells were regularly checked for mycoplasma contamination (MicoAlert Lonza).

### Intestinal monolayers and fatty acid supplementation

Monolayers were generated from intestinal organoids as described previously [35]. Briefly, established cat or mouse intestinal organoids were washed with cold PBS, digested by 0.05% m/v trypsin for 5 minutes at 37 ˚C, centrifuged at 250*g* for 3 minutes, and resuspended in fresh prewarmed organoid medium. Cell suspension was added into a chamber slide (Thermo) precoated with Entactin-Collagen IV-Laminin (Corning) for cat cells or 2% m/v Gelatin in PBS (Sigma) for mouse cells. The slides were coated by air-drying the basement membrane matrix or gelatin to air-dry overnight. The monolayers were grown for 10–15 days prior to infection with *T. gondii* bradyzoites, with media change every other day until cells reached 90% or more confluency. Linoleic acid or oleic acid conjugated to BSA (Sigma) was added to the organoid monolayers to 0.2 mM 24 hours prior to infection.

## Bradyzoite preparation and infection

C57BL/6J mice were oral gavage infected with 500–1,000 ME49 oocysts from cat feces. At 28 days post infection, brains were removed, washed in cold PBS, and homogenized with a glass tissue grinder. The suspension was centrifuged at 400*g* for 10 minutes and the pellet suspended in 20% m/v Dextran (Average MW 150,000, Sigma). Bradyzoite cysts were pelleted and separated from brain material by centrifugation at 2,200*g* for 10 minutes. The pellet was washed in PBS, digested by 0.1 mg/mL pepsin in HCl for 5 minutes at 37 ˚C, and then neutralized with an equal volume 1% Sodium Carbonate (Sigma). Bradyzoites were spun at 250*g* for 10 minutes, resuspended in prewarmed organoid medium, and added onto the organoid monolayers with a multiplicity of infection of 1 bradyzoite:10 intestinal epithelial cells (MOI 1:10).

## Delta-6-desaturase inhibition

SC-26196 (Cayman) was solubilized in DMSO and used at 20 µM in mouse organoid monolayers. For in vivo treatment, the inhibitor was solubilized in 0.5% m/v methylcellulose, and the mice were given 50 mg/kg every 12 hours by oral gavage [24]. C57BL/6J female mice (4 weeks old) deleted in Z-DNA-binding protein [36] were divided into four different groups: uninfected control, *T. gondii*–infected without fatty acid supplementation, *T. gondii*–infected with linoleic acid supplementation, and *T. gondii*–infected with linoleic acid and SC-26196 inhibitor. Each mouse supplemented with linoleic acid received 10 µL of 99% linoleic acid oil (MilliporeSigma Cat#843483) suspended in 0.5% Methylcellulose per day by oral gavage. Mice were infected with 1,000 brain cysts purified as described above by oral gavage and euthanized 7 days post infection. Sample size was at least two mice per group, and the experiment was repeated five times. Alternatively, each mouse was infected with one mouse brain at least 2 months post infection with at least 1,000 cysts. Mice were treated with SC-26196 until day 5 post infection. Feces were collected from days 5–14 and oocysts enumerated by microscopy.

## Immunofluorescence

Intestinal organoid monolayers or mouse fecal samples were fixed in 3.7% formaldehyde in PBS for 20 minutes, permeabilized with 0.2% triton X-100 (Sigma) in PBS at room temperature for 1 hour, and then blocked with 3% BSA in PBS at room temperature for 1 hour. Primary antibody was incubated at 4 ˚C overnight in 0.2% v/v Triton x-100 and 3% BSA in PBS (1:100 mouse anti-GRA11B, 1:100 rabbit anti-BRP1, 1:100 mouse anti-AO2, 1:50 monoclonal mouse anti-ZO-1 [Santa Cruz], or 1:25 mouse IgM anti-oocyst wall 3G4). Sporulated oocysts from mouse feces were dried onto slides, fixed and permeabilized with ice-cold acetone for 30 minutes, and incubated with 1:20 mouse 4B6 to the visualize the sporocyst. Slides were incubated with the specific secondary antibody (1:500 goat anti-rabbit Alexa Fluor 488 and 1:500 goat anti-mouse Alexa Fluor 594) at room temperature for 1 hour and then washed three times with PBS. Cells nuclei were stained with 10 µM DAPI (Sigma). Slides were mounted in Vectashield antifade mounting medium (VectorLabs). Samples were imaged on Zeiss Axioplan III equipped with a triple-pass (DAPI/fluorescein isothiocyanate [FITC]/Texas Red) emission cube, differential interference contrast optics, and a monochromatic Axiocam camera operated by Zen software (Zeiss) and processed using ImageJ (Fiji packet).

## Tissue sectioning and histology

Ileums were fixed in 3.7% formaldehyde in PBS overnight, embedded in paraffin, and sectioned by the Translational Research Initiatives in Pathology laboratory at the University of Wisconsin–Madison. The sections were stained with hematoxylin and eosin or reticulin (silver) stain.

## Real-time PCR on ileum cDNA

Mice with and without delta-6-desaturase inhibitor treatment were euthanized 7 days post infection. The ileum of each mouse was removed and homogenized in 1 mL of TRIzol. Total RNA was isolated according to manufacturer's protocol (Invitrogen) and treated with amplification-grade DNase I. cDNA was generated using the Invitrogen SuperScript III First-Strand Synthesis kit with random hexamer primers. GRA11B and SAG1 were used as markers of sexual and asexual stages, respectively. The *T. gondii* housekeeping gene TUB1A was used to normalize target gene expression. Real-time qPCR was performed using Bio-Rad iTaq Universal SYBR Green Supermix on an Applied Biosystems StepOnePlus Real-Time PCR system. The efficiency of each primer set was calculated from the slope of a 1:10 dilution standard curve of tachyzoite gDNA, where $E = 10^{\wedge}(-1/\text{slope})$. The Pfaffl method [37], which accounts for differences in efficiencies, was then used to calculate the relative gene expression of GRA11B and SAG1 per sample, in triplicate. Only wells with one melt curve temperature were used, indicating a single product. Primer sequences were as follows:

TUB1A forward: 5′ –GACGACGCCTTCAACACCTTCTTT– 3′

Reverse: 5′ –AGTTGTTCGCAGCATCCTCTTTCC– 3′

SAG1 forward: 5′ –TGCCCAGCGGGTACTACAAG– 3′

Reverse: 5′ –TGCCGTGTCGAGACTAGCAG– 3′

GRA11B forward: 5′ –ATCAAGTCGCACGAGACGCC– 3′

Reverse: 5′ –AGCGAATTGCGTTCCCTGCT– 3′

## Real-time PCR for *T. gondii* genomic DNA in fecal samples

Fecal samples from the mice with and without delta-6-desaturase inhibitor treatment were collected. Genomic DNA was generated from 0.1 g of feces from each mouse using the power soil DNA kit (QIAGEN) according to the manufacturer's instructions except that cells were broken by a bead beater instead of a vortex. A standard curve was generated using a dilution series of $10^1$–$10^5$ parasites per well amplified using the SAG1 primer set described above, based on a genomic DNA sample with known parasite quantity. The Ct values were plotted against the log of the parasite numbers. The number of target gene copies in each sample can be interpolated from the linear regression of the standard curve.

$$\text{target gene copy \#} = 10^{\wedge \frac{(\text{Target gene Ct} - \text{y intercept})}{\text{slope}}}$$

Real-time PCR was performed on each sample, in triplicate, using Bio-Rad iTaq Universal SYBR Green Supermix on an Applied Biosystems StepOnePlus Real-Time PCR system. The calculated copy numbers of each sample were normalized based on the ng of nucleic acid used as PCR template. Only wells with one melt curve temperature were used, indicating a single product.

## PCR of cat intestinal monolayers

Cat intestinal monolayers were grown in 24-well plates until confluency and then were incubated with either no fatty acid supplementation, 200 μM oleic acid, or 200 μM linoleic acid for 24 hours. The monolayers were infected with ME49 bradyzoites purified from brains of chronic infected mice in duplicate with uninfected monolayers as a negative control. Seven days post infection, RNA was extracted with TRIzol, and cDNA was synthesized as described

above. TgAO2 was used as a marker for macrogametes, and TgME49_306338 was used as a marker for microgametes. TUB1A was used as an input control using the same primers as above. A cDNA synthesis reaction without the addition of reverse transcriptase was used as a control for genomic DNA contamination. Equivalent amounts of cDNA per sample were used as a template for each PCR reaction, and the products were separated on an acrylamide gel and imaged. Primer sequences were as follows:

TgAO2 forward: 5′ –GTCTTGGTTCGTTGAAGGGGCTG– 3′

Reverse: 5′ –CGTCCTCGATGCCCATGAAATCTG– 3′

TgME49_306338 forward: 5′ –CCACGTCCTTCGCCGATG– 3′

Reverse: 5′ –CATCAGAGGTCCCAGGTTGTCG– 3′

## Statistical methods

All real-time PCR fecal samples were run in triplicate technical replicates. The difference between the mean target gene copy numbers was analyzed by two-tailed unpaired *t* tests. The real-time PCR intestinal samples were run in triplicate from two biological replicates per group. The difference between the mean relative expression of each target gene was analyzed by two-tailed unpaired *t* tests.

## Oocyst sporulation and mouse infections

Fresh fecal samples were obtained from each mouse, homogenized in PBS, and then centrifuged at 1,500*g*. The pellet was resuspended in PBS plus penicillin and streptomycin, and the samples were shaken for 7–14 days at room temperature in presence of oxygen. Mice oocysts were stable for at least 3 months at 4 ˚C. Naïve mice were infected with approximately 250 mouse oocysts through intraperitoneal injection. Mice were humanely euthanized at day 28 post infection, and their brains were removed, homogenized, and either incubated with biotinylated DBA 1:1,000 or purified with 20% m/v Dextran as described above before DBA incubation. All cysts were then incubated with streptavidin-conjugated Alexa Fluor-594 1:1,000 and imaged on Zeiss Axioplan III equipped with a triple-pass (DAPI/FITC/Texas Red) emission cube, differential interference contrast optics, and a monochromatic Axiocam camera operated by Zen software (Zeiss) and processed using ImageJ (Fiji packet).

## Western immunoblot

ME49 tachyzoite lysates were run on a 15% SDS-PAGE protein gel, transferred to a nitrocellulose membrane, and strips blocked with 5% w/v low fat milk in TBS 1.0% v/v Tween-20. Collected serum was diluted 1:250 TBS 1.0% v/v Tween-20, and 1:2,000 anti-mouse HRP was used as the secondary antibody. Serum from C57BL/6 mice infected with cat oocysts for 26 days was used as positive control, and serum from uninfected C57BL/6 mice was used as negative control. Stripes were imaged by LI-COR (LI-COR Biosciences) at 700 nm or chemiluminescence for a 5-minute exposure.

## Supporting information

**S1 Fig. Quantification of merozoites in mouse tissue culture.** Mouse intestinal organoids were disassociated by trypsin, and the individual cells were grown on glass slides. Slides were divided into three different groups: not supplemented with fatty acids or SC-26196 inhibitor, supplemented with 200 μM linoleic acid, or supplemented with 200 μM linoleic acid plus

20 μM SC-26196 ("D6D +LA"). Monolayers were infected with *T. gondii* ME49 bradyzoites purified from brains of chronic infected mice. At 5 days post infection, monolayers were stained for GRA11B, BRP1, and DAPI. Total number of parasitophorous vacuoles were counted by positive DAPI staining and confirmed by morphology with DIC. The percentage of vacuoles positive for GRA11B and BRP1 out of the total vacuoles was determined. Three biological replicates were counted, and on average, 26% of the total vacuoles were positive for GRA11B and BRP1 in the linoleic acid–supplemented monolayers with the addition of SC-26196. $**p$-value = 0.0039 with $N = 3$ by two-tailed unpaired $t$ test. BRP1, bradyzoite rhoptry protein-1; DIC, differential interference contrast; GRA11B, dense granule protein 11B.
(TIF)

**S2 Fig. Mice oocyst infectivity evaluation by serum conversion.** Oocysts collected from SC-26196-treated mice were sporulated and injected intraperitonially into C57BL/6 mice. Serum was collected by terminal bleed at either 22 days (1 and 2), 28 days (3 and 4), or 3 months (5–10) post infection. Serum from uninfected mouse was used as a negative control (−). The positive control (+) was serum from a mouse infected for 26 days with cat oocysts. Serum was incubated with nitrocellulose blotted with ME49 tachyzoite lysate. Panel (a) is chemiluminescence, and (b) is a 700-nm image showing the individual lanes and the protein ladder.
(TIF)

**S3 Fig. Mouse oocysts are infectious.** Twenty-eight days post infection with oocysts, mouse brains were removed, homogenized, and stained for *T. gondii* cysts by DBA (red). All panels are 20 μm$^2$ with a 5-μm white size bar in the lower right corner. DBA, *D. biflorus* agglutinin.
(TIF)

**S1 Data. Raw data for Fig 2a.** Quantification of GRA11B and BRP1 double-positive vacuoles in cat tissue culture. Cat intestinal monolayers were divided into three different groups: not supplemented with fatty acid, supplemented with 200 μM oleic acid, or supplemented with 200 μM linoleic acid (left column). Monolayers were infected with *T. gondii* ME49 bradyzoites purified from brains of chronic infected mice at a 1:10 MOI. Five days after infection, staining was performed for GRA11B and BRP1 along with DAPI. For each random field, the number of host cell nuclei was counted along with GRA11B/BRP1 double-positive and negative vacuoles. For each biological replicate, five different technical replicates were counted, and the average value is presented. Each line of the table is an independent experiment, three biological replicates. BRP1, bradyzoite rhoptry protein-1; GRA11B, dense granule protein 11B; MOI, multiplicity of infection.
(TIF)

**S2 Data. Raw data for Fig 2b.** Raw Ct values of TUB1A, SAG1, and GRA11B from the cDNA of cat intestinal monolayers samples using TUB1A as the normalizer for target gene expression. Wells with multiple melt curve temperatures, indicating off-target products, were excluded ("NA"). Samples below the detection limit of 40 cycles are labeled BDL. BRP1, bradyzoite rhoptry protein-1; GRA11B, dense granule protein 11B; SAG1, surface antigen 1; TUB1A, tubulin 1A.
(TIF)

**S3 Data. Raw data for Fig 3h.** Quantification of vacuoles positive with the oocyst wall antibody 3G4 in cat tissue culture. Cat intestinal monolayers were divided into three different groups: not supplemented with fatty acid, supplemented with 200 μM oleic acid, or supplemented with 200 μM linoleic acid (left column). Monolayers were infected with *T. gondii*

ME49 bradyzoites purified from brains of chronic infected mice at a 1:10 MOI. Seven days after infection, staining for oocyst wall antigen was performed with 3G4 antibody. For each 1-cm$^2$ well, the total number of 3G4 positive vacuoles was counted. Each column of the table is an independent experiment, three biological replicates. MOI, multiplicity of infection.
(TIF)

**S4 Data. Raw data for S1 Fig.** Quantification of GRA11B and BRP1 double-positive vacuoles in mouse tissue culture. Mouse intestinal monolayers were divided into three different groups: not supplemented with fatty acids or SC-26196 inhibitor (no fatty acid), supplemented with 200 μM linoleic acid, or supplemented with 200 μM linoleic acid plus the addition of 20 μM SC-26196 ("D6D inh."). Monolayers were infected with *T. gondii* ME49 bradyzoites purified from brains of chronic infected mice at a 1:10 MOI. Five days after infection, staining was performed for GRA11B and BRP1 along with DAPI. For each random field, the number of host cell nuclei was counted along with GRA11B/BRP1 double-positive and negative vacuoles. For each biological replicate, three different technical replicates were counted, and the average value is presented. Each line of the table is an independent experiment, three biological replicates. BRP1, bradyzoite rhoptry protein-1; GRA11B, dense granule protein 11B; MOI, multiplicity of infection.
(TIF)

**S5 Data. Raw data for Fig 5a.** Raw Ct values of the TUB1A, SAG1, and GRA11B standard curves on a dilution series of genomic DNA from tachyzoites and raw Ct values of TUB1A, SAG1, and GRA11B from the cDNA of homogenized mouse ileum samples using TUB1A as the normalizer for target gene expression. Wells with multiple melt curve temperatures, indicating off-target products, were excluded ("NA"). Samples below the detection limit of 40 cycles are labeled BDL. BRP1, bradyzoite rhoptry protein-1; GRA11B, dense granule protein 11B; SAG1, surface antigen 1; TUB1A, tubulin 1A.
(TIF)

**S6 Data. Raw data for Fig 6a.** Raw Ct values of the SAG1 standard curve on a dilution series of genomic DNA with known parasite quantity and raw Ct values of SAG1 amplification from genomic DNA of unknown fecal samples using ng quantity input as the normalizer. Wells with multiple melt curve temperatures, indicating off-target products, were excluded ("NA"). SAG1, surface antigen 1.
(TIF)

**S7 Data. Raw data for Fig 6b.** Feces (0.5 g) was solubilized in 500 μL of saline, and then 10 μL of this suspension was counted in a hemocytometer. The values in the table are number of oocysts per sample. Each column of the table is an independent experiment, three biological replicates. The average counts per d.p.i. were used to calculate the number of oocysts/gram of feces. d.p.i., day post infection.
(TIF)

# Acknowledgments

We sincerely thank Jason Spence and his lab for assistance with intestinal organoid culture; Aurélien Dumètre, Adrian Hehl, and John Boothroyd for cat stage-specific antibodies; Maria Arendt for assistance with intestinal pathology images; Heather Fritz, David Ferguson, and Jean François Dubremetz for advice; and Christina Hull, Benjamin Rosenthal, and Rodney Welch for editing of the manuscript.

## Author Contributions

**Conceptualization:** Bruno Martorelli Di Genova, Laura J. Knoll.

**Formal analysis:** Bruno Martorelli Di Genova, Sarah K. Wilson.

**Funding acquisition:** Laura J. Knoll.

**Investigation:** Bruno Martorelli Di Genova, Sarah K. Wilson, Laura J. Knoll.

**Methodology:** Bruno Martorelli Di Genova, Sarah K. Wilson, J. P. Dubey, Laura J. Knoll.

**Project administration:** Laura J. Knoll.

**Resources:** Laura J. Knoll.

**Supervision:** Laura J. Knoll.

**Validation:** Bruno Martorelli Di Genova, Sarah K. Wilson, J. P. Dubey, Laura J. Knoll.

**Visualization:** Bruno Martorelli Di Genova.

**Writing – original draft:** Bruno Martorelli Di Genova, Laura J. Knoll.

**Writing – review & editing:** Bruno Martorelli Di Genova, Sarah K. Wilson, J. P. Dubey, Laura J. Knoll.

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
