## [Editor Report · Decision Letter 0]

20 Jun 2019

Dear Laura, 

Thank you for submitting your manuscript entitled "Intestinal delta-6-desaturase activity determines host range for Toxoplasma sexual reproduction" for consideration as a Research Article by PLOS Biology following our chat yesterday.

I've evaluated your manuscript and consulted with an academic editor with relevant expertise and I am writing to let you know that we would like to send your submission out for external peer review.

**Important**: Please also see below for further information regarding completing the MDAR reporting checklist. The checklist can be accessed here: https://plos.io/MDARChecklist

Please re-submit your manuscript and the checklist, within two working days, i.e. by Jun 22 2019 11:59PM.

Kind regards,

Lauren

Lauren A Richardson, Ph.D

Senior Editor

PLOS Biology

INFORMATION REGARDING THE REPORTING CHECKLIST:

PLOS Biology is pleased to support the "minimum reporting standards in the life sciences" initiative (https://osf.io/preprints/metaarxiv/9sm4x/). This effort brings together a number of leading journals and reproducibility experts to develop minimum expectations for reporting information about Materials (including data and code), Design, Analysis and Reporting (MDAR) in published papers. We believe broad alignment on these standards will be to the benefit of authors, reviewers, journals and the wider research community and will help drive better practise in publishing reproducible research. 

We are therefore participating in a community pilot involving a small number of life science journals to test the MDAR checklist. The checklist is intended to help authors, reviewers and editors adopt and implement the minimum reporting framework. 

IMPORTANT: We have chosen your manuscript to participate in this trial. The relevant documents can be located here:

MDAR reporting checklist (to be filled in by you): https://plos.io/MDARChecklist

**We strongly encourage you to complete the MDAR reporting checklist and return it to us with your full submission, as described above. We would also be very grateful if you could complete this author survey:

https://forms.gle/seEgCrDtM6GLKFGQA

Additional background information:

Interpreting the MDAR Framework: https://plos.io/MDARFramework

Please note that your completed checklist and survey will be shared with the minimum reporting standards working group. However, the working group will not be provided with access to the manuscript or any other confidential information including author identities, manuscript titles or abstracts. Feedback from this process will be used to consider next steps, which might include revisions to the content of the checklist. Data and materials from this initial trial will be publicly shared in September 2019. Data will only be provided in aggregate form and will not be parsed by individual article or by journal, so as to respect the confidentiality of responses. 

Please treat the checklist and elaboration as confidential as public release is planned for September 2019.

We would be grateful for any feedback you may have.

---

## [Decision Letter · Decision Letter 1]

8 Jul 2019

Dear Dr Knoll,

Thank you very much for submitting your manuscript "Intestinal delta-6-desaturase activity determines host range for Toxoplasma sexual reproduction" for consideration as a Research Article by PLOS Biology. Your paper was evaluated by the PLOS Biology editors as well as by an Academic Editor with relevant expertise and by three independent reviewers. The reviewers appreciated the attention to an important topic. Based on the reviews, we will probably accept this manuscript for publication, providing that you will modify the manuscript according to the review recommendations.

We expect to receive your revised manuscript within two weeks. Your revisions should address the specific points made by each reviewer. In addition to the remaining revisions and before we will be able to formally accept your manuscript and consider it "in press", we also need to ensure that your article conforms to our guidelines, one of which is described below under DATA POLICY. A member of our team will be in touch shortly with a set of requests. As we can't proceed until these requirements are met, your swift response will help prevent delays to publication.

Please note that you may have the opportunity to make the peer review history publicly available. The record will include editor decision letters (with reviews) and your responses to reviewer comments. If eligible, we will contact you to opt in or out.

Sincerely,

Di Jiang, PhD

Associate Editor

on behalf of

Lauren A Richardson, Ph.D, 

Senior Editor

PLOS Biology

DATA POLICY:

Regardless of the method selected, please ensure that you provide the individual numerical values that underlie the summary data displayed in the following figure panels: Figures 2ab, 3h, 5a, 6ab, S1, as they are essential for readers to assess your analysis and to reproduce it. Please also ensure that figure legends in your manuscript include information on where the underlying data can be found.

Reviewer remarks:

Reviewer #1: Summary

The work described in this manuscript is truly transformative as it not only identifies the trigger of the Toxoplasma sexual cycle and why this only occurs in cats, it continues to fulfil Koch’s postulates by overcoming this species barrier by demonstrating the completion of the cycle in mice (and largely in vitro as well) by mimicking the cat environment. The work is also transformative in its approach as this is the first application of cat intestinal organoids to Toxoplasma. Besides the major biological breakthrough, these insights are relevant toward developing sexual recombination experiments in Toxoplasma (a tool that has been cumbersome due to its poor efficiency and the need of cats) as well as toward developing control strategies of cats shedding oocysts, which is a responsible for a large portion of human infections with Toxoplasma. In addition, the work is timely in light of the recent sudden and politically motivated suspension of the use of cats at the USDA, which supplied the majority of the Toxoplasma field with Toxoplasma oocysts. This work is therefore relevant for a very wide audience of microbiologists, infectious disease specialists, clinicians, epidemiologists and basic parasitology researchers. By fulfilling Koch’s postulates the paper per definition is very rigorous. In summary, a through breakthrough manuscript well described and executed. Only one minor comment as described below.

Specific comments.

Supplementary information; No Fig S2, but a fig 6. This is likely a mix up. Legend says “strained” instead of “stained” for DBA staining.

P7 line 13: in the sentence is referred to sero conversion, however, no data supporting this statement are shown in neither Fig 6e nor S2.

Reviewer #2 (Felix Yarovinsky, signed review): In this groundbreaking work the authors succeeded in answering a major biological question in the field of parasite biology. It was a mystery why the sexual cycle of Toxoplasma gondii is limited to felines, while the asexual life cycle can happen in any warm-blooded mammal. Using an elegant in vitro system based on the intestinal organoids, the authors identified that linoleic acid is essential for the expression of markers associated with the sexual cycle of the parasite. The authors provide the detailed biochemical explanation for this requirement. It appears that cats are deficient in delta-6-desaturase, the rate‐limiting step for the conversion of linoleic acid to arachidonic acid. As a result, linoleic acid is the dominant fatty acid in cat serum. This explains why the felines have an increased level of linoleic acid. 

The authors next validated in vitro observation with mice that were fed with linoleic acid and delta-6-desaturase inhibitor. This treatment results in the full sexual cycle of the parasite in the murine intestine. This is a truly remarkable discovery in biology, and while there might be minor technical gaps in the work, I believe an immediate publication would benefit the scientific community. This is a highly significant work that not only teach us new biology, but also has a major clinical significance since it provides a novel biochemical tool to deal with toxoplasmosis in cats that may eventually lead to the elimination of this parasite in cats. 

I only have two minor comments 

1. In the following statement ‘After 5 days of infection, we found that the addition of linoleic acid but not oleic acid caused approximately 35% of the T. gondii to express both merozoite stage markers (Fig. 2a)’ the authors forgot to acknowledge the data shown in Fig 1d-1g as an additional experimental support in the above-mentioned statement. 

Figure 2, Y- axis: the labeling needs to be more specific, as ‘%GRA11B and BRP1 positive’ is confusing.

Reviewer #3: Although it has been known for decades that felids are the definitive host for sexual replication of the intracellular protozoan parasite Toxoplasma gondii, the molecular underpinnings of this remarkable host specificity for parasite mating has remained an important unresolved mystery. The current work compellingly establishes that an excess linoleic acid drives sexual differentiation in culture and in mice, thus establishing for the first time in vitro and alternative in vivo models for T. gondii sexual development. The work represents a key breakthrough that will permit further dissection of sexual development, accelerate classical genetic linkage mapping of important traits, and reduce the dependence on using cats as an experimental model, among other avenues. The authors provide thorough qualitative and quantitative data that is appropriately interpreted and supportive of the conclusions. The following minor considerations should be addressed.

Line 21. Delete on of the “withs”

Page 8, line 5-7. This statement should be softened from “likely” given that no data is presented to support it. It is suggested that the authors rephrase it to e.g., It is possible that the multiple…

---

## [Editor Report · Decision Letter 2]

17 Jul 2019

Dear Dr Knoll,

On behalf of my colleagues and the Academic Editor, Boris Striepen, I am pleased to inform you that we will be delighted to publish your Research Article in PLOS Biology. 

Early Version

PRESS 

Kind regards,

Harry Porter

Publication Assistant, 

PLOS Biology

on behalf of

Lauren Richardson,

Senior Editor

PLOS Biology